# Early-Stage Neural Network Hardware Performance Analysis

**Alex Karbachevsky** [1,†]**, Chaim Baskin** [1,*,†]**, Evgenii Zheltonozhskii** [1,†]**, Yevgeny Yermolin** [1]**, Freddy Gabbay** [2]**, Alex M. Bronstein** [1] **and Avi Mendelson** [1]

1 Technion—Israel Institute of Technology, Haifa 3200003, Israel; alex.k@campus.technion.ac.il (A.K.); evgeniizh@campus.technion.ac.il (E.Z.); yevgeny.ye@cs.technion.ac.il (Y.Y.); bron@cs.technion.ac.il (A.M.B.); avi.mendelson@cs.technion.ac.il (A.M.)
2 Ruppin Academic Center, Emek Hefer 4025000, Israel; freddyg@ruppin.ac.il
* Correspondence: chaimbaskin@campus.technion.ac.il
† These authors contributed equally to this work.

**Abstract:** The demand for running NNs in embedded environments has increased significantly in recent years due to the significant success of convolutional neural network (CNN) approaches in various tasks, including image recognition and generation. The task of achieving high accuracy on resource-restricted devices, however, is still considered to be challenging, which is mainly due to the vast number of design parameters that need to be balanced. While the quantization of CNN parameters leads to a reduction of power and area, it can also generate unexpected changes in the balance between communication and computation. This change is hard to evaluate, and the lack of balance may lead to lower utilization of either memory bandwidth or computational resources, thereby reducing performance. This paper introduces a hardware performance analysis framework for identifying bottlenecks in the early stages of CNN hardware design. We demonstrate how the proposed method can help in evaluating different architecture alternatives of resource-restricted CNN accelerators (e.g., part of real-time embedded systems) early in design stages and, thus, prevent making design mistakes.

**Keywords:** neural networks; accelerators; quantization; CNN architecture

## 1. Introduction

Many domain-specific systems have been found to be efficient, in particular, when developing low resource devices, for example, for IoT applications. A system architect designing such devices must consider hardware limitations (e.g., bandwidth and local memory capacity), algorithmic factors (e.g., accuracy and representation of data), and system aspects (e.g., cost, power envelop, and battery life). Many IoT and other resource-constrained devices provide support for applications that use convolutional neural networks (CNNs). CNNs can achieve spectacular performance in various tasks that cover a wide range of domains, such as computer vision, medicine, autonomous vehicles, robotics, etc. Notwithstanding, CNNs contain a vast number of parameters and they require a significant amount of computation during inference, thus monopolizing hardware resources and demanding massively parallel computation engines. These requirements have led to great interest in using custom-designed hardware for the efficient inference of CNNs. For example, such hardware allows for neural networks (NNs) to be used in real-life applications, such as real-time monitoring system for human activities [1], autonomous laparoscopic robotic surgery [2–4], or deployed on low-power edge devices or as part of an IP in an SoC. Developing energy efficient CNN accelerators requires a new set of design tools, due to the tight entanglement between the algorithmic aspects, the chip architecture, and the constraints that the end product needs to meet. Great efforts have indeed already been made for developing low-resource CNN architectures [5–8].

The splitting of the regular 3 × 3 convolutions into a channel-wise 3 × 3 convolution, followed by a 1 × 1 one, is one example of architectural changes. Another way to reduce

the computational burden is to quantize the CNN parameters (weights and activations), employing low-bit integer representation of the data instead of expensive floating-point representation. Recent quantization-aware training schemes [9–13] achieve near-baseline accuracy for as low as 2-bit quantization. Quantizing the CNN parameters reduces both the number of gates required for each multiply-accumulate (MAC) operation and the amount of routing. In addition, quantization reduces the bandwidth requirements for external and internal memory. The architect needs to make fundamental decisions early in the design process (accuracy requirements, their impact on performance and power, and vis-á-vis which algorithm is going to be used) and no existing tool can help predict the effect of these design factors ahead of time. If the CNN is to be an asset to users, the impact of the accelerator's high-level architecture (e.g., the amount of layers, their size, and the bitwidth of the operands), on the power, the area, and the performance of the final product needs to be defined and predicted at an early stage of the project.

Recent research has shown that ASIC-based architectures are the most efficient solution for CNN accelerators in both datacenters [14–19] and real-time platforms [20–22]. Accordingly, we employ an implementation of a streaming [23] ASIC-based convolutional engine for our experiments. Nevertheless, our methodology can be applied when evaluating other types of architectures, such as FPGA-based accelerators [24–26]. In both cases, the development process includes an important trade-off between the logical gates area, local memory area, and their routing of the accelerator design versus the performance and accuracy of the resulting system. This is especially true in an SoC IC, where the CNN accelerator is a small part of the entire system, and the remaining silicon "budget" needs to be divided between execution units and local memory; here, these trade-offs have great impact. Moreover, all of these parameters also depend on the parameters' quantization level, and its impact on both communication and computation.

To date, there is no quantitative metric for this trade-off available at the CNN accelerator design stage and no tool exists that can assist the architect in predicting the impact of high level decisions on the important design implementation parameters. Ideally, the designer would like to have an early estimation of the chip resources that are required by the accelerator, as well as the performance, accuracy, and power that it can achieve. A critical difficulty in trying to predict the design parameters for CNN-based systems is the lack of a proper complexity metric. Currently, the most common metric for calculating the computational complexity of CNN algorithms is the number of MAC operations, denoted as OPS (or FLOPS in case of floating-point operations). However, this metric does not take into account the data format or additional operations that are performed during the inference, such as memory accesses and communication. For this reason, the number of FLOPS does not necessarily correlate with run-time [27,28] or the required amount of computational resources.

This paper proposes an adapted roofline analysis tool in order to accommodate variable bitwidth configurations, and it views the entire network computation and communication needs on a single plot. We show how this analysis framework can be used in order in the early stages of an accelerator design to analyze the trade-off between the number of processing engines (PEs) and the quantization of parameters. Moreover, we demonstrate the performance analysis of CNN architecture (VGG-16) on an existing accelerator.

### 1.1. Contribution

This paper makes several contributions. Firstly, we study the impact of CNN quantization on the hardware implementation of computational resources.

Secondly, we extend the previously proposed computation complexity for quantized CNNs, termed BOPS [29], while using a communication complexity analysis. We assist in identifying the performance bottlenecks that may arise from the data movement by extending the roofline model [30]. We also demonstrate how the proposed tool can be used to assist architecture-level decisions in the early design stages.

Finally, we demonstrate the efficiency of performance bottlenecks analysis while using the proposed method on a basic quantized convolution accelerator that we have created for this purpose, and on an existing machine learning hardware accelerator, Eyeris [20].

### 1.2. Related Work

In this section, we provide an overview of prior work that proposed metrics for estimating the complexity and power/energy consumption of different workloads, while focusing on CNNs. FLOPS is the most commonly used metric for evaluating computational complexity [31]: the amount of floating-point operations required to perform the computation. In the case of integer operations, the obvious generalization of FLOPS is OPS, which is just the number of operations (not necessarily floating point). A fundamental limitation of these metrics is the assumption that the same data representation is used for all operations; otherwise, the calculated complexity does not reflect the real hardware resource complexity. Wang et al. [32] claimed that FLOPS is an inappropriate metric for estimating the performance of workloads executed in datacenters and proposed a basic operations metric that uses a roofline-based model, while taking the computational and communication bottlenecks into account for a more accurate estimation of the total performance. Sze et al. [27] described the factors affecting efficiency and, in particular, showed that the use of a peak performance metric (such as TOPS/W) is not enough. Rather, they asserted that measuring efficiency requires multiple metrics, including accuracy, throughput, latency, energy consumption, power consumption, cost, flexibility, and scalability.

Parashar et al. [33] provided a system-level evaluation and exploration of architectural attributes of CNN accelerators and a broad range of hardware typologies. They generated an accurate estimation of performance and power while using a mapper that explores the many ways to schedule the work on individual PEs to find the optimal solution. By describing the accelerator structure, which includes compute elements, external and local memory hierarchy, and the network between them, they determine the performance for any network structure and, by utilizing Accelergy [34], the power usage. However, this system level approach lacks the fine granularity of quantization.

In addition to general-purpose metrics, other metrics were specifically developed for an evaluation of NN complexity. Mishra et al. [35] defined the "compute cost" as the product of the number of fused multiply-add (FMA) operations and the sum of the width of the activation and weight operands, without distinguishing between floating- and fixed-point operations. While using this metric, the authors claimed to have reached a $32\times$ "compute cost" reduction by switching from FP32 to binary representation. Still, as we further show on in our paper, this drop is not the real reduction in the hardware components. Jiang et al. [36] noted that a single metric cannot comprehensively reflect the performance of deep learning (DL) accelerators. They investigate the impact of various frequently-used hardware optimizations on a typical DL accelerator and then quantify their effects on the accuracy and throughput on under-representative DL inference workloads. Their major conclusion is that hardware throughput is not necessarily correlated with the end-to-end inference throughput of data feeding between host CPUs and AI accelerators. Finally, Baskin et al. [29] proposed generalizing FLOPS and OPS by taking into account the bitwidth of each operand as well as the operation type. The resulting metric, named BOPS (binary operations), allows for the area estimation of quantized CNNs, including cases of mixed quantization. The shortcoming of the aforementioned metrics is that they do not provide any insight on the amount of silicon resources needed to implement them. Our work, accordingly, functions as a bridge between the CNN workload complexity and the real power and area estimation.

## 2. Method

In this section, we describe our analysis framework. The framework provides the tools for analyzing performance bottlenecks on quantized NNs with arbitrary bitwidth for weights and activations, and a method for the accurate estimation of the silicon area in the early stage of the design. We start by describing the impact on the silicon area when switching from a floating-point representation to a fixed-point one. Subsequently, we present our area estimation approach, which assesses three elements: the computational complexity of the data path while using BOPS, which quantifies the hardware resources needed to implement the CNN computation engine on the silicon; the amount of local SRAM, which affects the area budget on the silicon; and the communication complexity, which defines the memory access pattern and bandwidth. Lastly, we present a modified roofline analysis tool for evaluating the performance bottlenecks of quantized networks. The fixed-point multiplication results that are presented in this section are based on the Synopsys standard library multiplier while using TSMC's 28 nm process.

### 2.1. The Impact of Quantization on Hardware Implementation

Currently, the most common representation of weights and activations for training and inference of CNNs is either 32-bit or 16-bit floating-point numbers. However, the fixed-point MAC operation requires significantly fewer hardware resources, even for the same input bitwidth. In order to illustrate this fact, we generated two multipliers: one for 32-bit floating-point operands (FPU100 from https://opencores.org/projects/fpu100) and the other for 32-bit fixed-point operands. The results presented in Table 1 show that a fixed-point multiplier uses approximately eight times fewer resources (area, gates, and power) than the floating-point counterpart. Next, we generated a PE that calculates a convolution with a $3 \times 3$ kernel, a basic operation in CNNs consisting of $3 \times 3 = 9$ MAC operations per output value. After switching from floating-point to fixed-point, we studied the area of a single PE with variable bitwidth. Note that the accumulator size depends on the network architecture: the maximal bitwidth of the output value is $b_w b_a + \log_2(k^2) + \log_2(n)$, where $n$ is the number of input features. Because extreme values are very rare, it is often possible to reduce the accumulator width without harming the network's accuracy [16].

**Table 1.** Key characteristics of 32-bit floating-point and 32-bit fixed-point multiplier designs. The fixed-point multiplier uses approximately eight times less area, gates, and power than the floating-point one.

| Multiplier | Gates | Cells | Area [μm²] | Power [mW] | | | |
|---|---|---|---|---|---|---|---|
| | | | | Internal | Switching | Leakage | Dynamic |
| Floating-Point | 40,090 | 17,175 | 11,786 | 2.76 | 1.31 | 0.43 | 10.53 |
| Fixed-Point | 5065 | 1726 | 1489 | 0.49 | 0.32 | 0.04 | 1.053 |

Figure 1 shows the silicon area of the PE as a function of the bitwidth; the layout of the computation engine we built for this paper is shown in Figure 2. We performed a polynomial regression and observed that the PE area had a quadratic dependence on the bitwidth, with the coefficient of determination $R^2 = 0.9999877$. This nonlinear dependency between the PE's area and operands' bitwidth demonstrates that the quantization impact on a network's hardware resources is quadratic: reducing the bitwidth of the operands by half reduces the area and, by proxy, the power by approximately a factor of four (contrary to what is assumed by, e.g., Mishra et al. [35]).

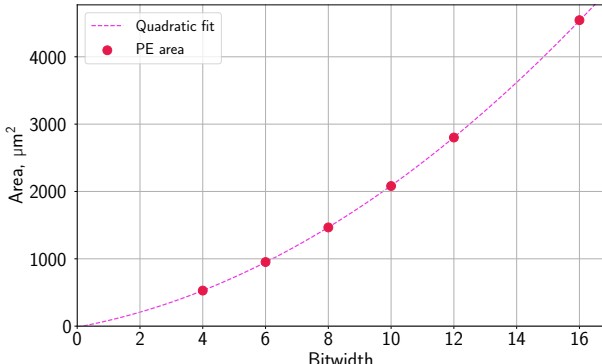

**Figure 1.** Area vs. bitwidth for a $3 \times 3$ PE with a single input and output channel. All of the weights and activations use the same bitwidth and the accumulator width is four bits larger, which is enough to store the result. The quadratic fit is $A = 12.39b^2 + 86.07b - 14.02$ with a goodness of fit $R^2 = 0.9999877$, where $A$ is the area and $b$ is the bitwidth of the PE.

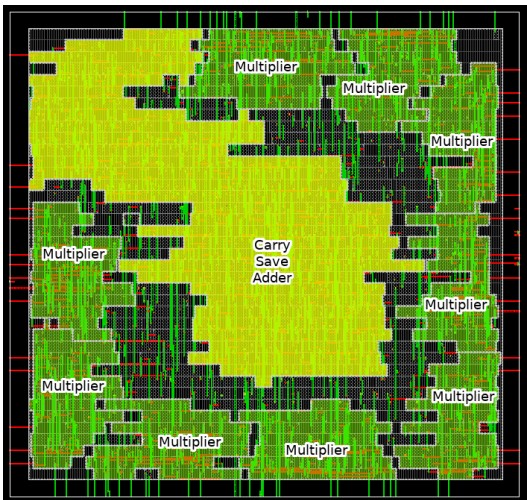

**Figure 2.** Our $3 \times 3$ kernel 8-bit processing engine (PE) layout using TSMC 28 nm technology. The carry-save adder can fit 12-bit numbers, which is large enough to store the output of the convolution.

### 2.2. Data Path

We now present the BOPS metric that is defined in Baskin et al. [29] as our computation complexity metric for the data path circuit. In particular, we show that BOPS can be used as an estimator for the area of the PEs in an accelerator. The area, in turn, is found to be linearly related to the power in case of the PEs.

The computation complexity metric describes the amount of arithmetic "work" that is needed to calculate the entire network or a single layer. BOPS is defined as the number of bit operations required to perform the calculation: the multiplication of $n$-bit number by $m$-bit number requires $n \cdot m$ bit operations, while addition requires $\max(n, m)$ bit operations. In particular, Baskin et al. [29] showed that a $k \times k$ convolutional layer with $b_a$-bit activations and $b_w$-bit weights requires

$$\text{BOPS} = mnk^2 \left( b_a b_w + b_a + b_w + \log_2(nk^2) \right) \tag{1}$$

bit operations, where $n$ and $m$ are, respectively, the number of input and output features of the layer. This definition takes the width of the accumulator required to accommodate the intermediate calculations into account, which depends on $n$. The BOPS of an entire network is calculated as a sum of the BOPS of the individual layers.

In Figure 3, we calculated BOPS values for the PE design shown in Figure 1 and added additional data points for the bitwidth of the weights and activation, including mixed precision between the two and then plotted them against the area. We conclude that, for a single PE with variable bitwidth, BOPS can be used in order to predict the PE area with high accuracy.

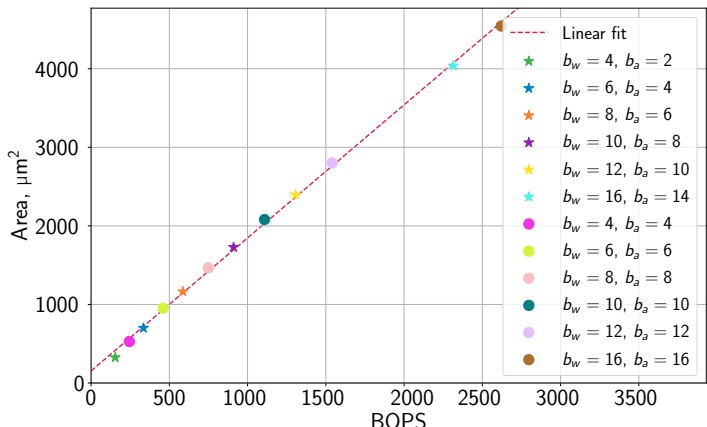

**Figure 3.** Area vs. BOPS for a $3 \times 3$ PE with a single input and output channel and variable bitwidth. The linear fit is $A = 1.694B + 153.46$ with a goodness of fit $R^2 = 0.998$, where $A$ is the area and $B$ is BOPS.

Next, we tested the predictive power of BOPS scaling with the size of the design. We generated several designs with variable bitwidths, $b_w = b_a \in \{4, 6, 8\}$, and variable numbers of PEs $n = m \in \{4, 8, 16\}$, connected together in a "weight stationary" architecture, in which a set of weights is loaded and then kept in local memory, while the activations are read from main memory, used to accommodate multidimensional inputs and outputs that typically arise in real CNN layers. Figure 4 shows that the area linearly depends on the BOPS for the range of two orders of magnitude of total area with goodness of fit $R^2 = 0.998$. We conclude that BOPS provides a high-accuracy approximation of the area and power required by the hardware and, thus, can be used as an estimator in early design stages. While the area of the accelerator depends on the particular design of the PE, this only affects the slope of the linear fit, since the area is still linearly dependent on the amount of PEs. An architect can use high level parameters, such as the number of input features and output features, kernel size, etc., to obtain an early estimation of how much power and silicon area are needed to run the network, without having any knowledge regarding VLSI constraints.

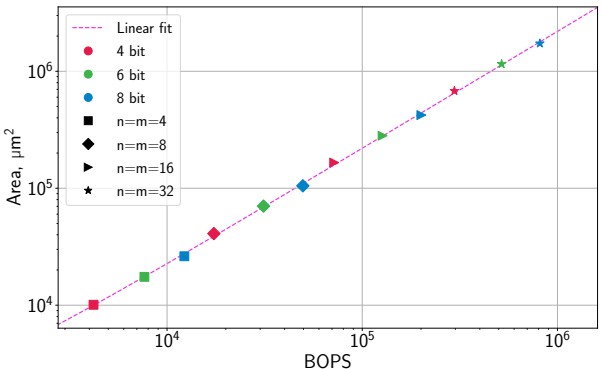

**Figure 4.** Area vs. BOPS for a $3 \times 3$ PE with variable input (n) and output (m) feature dimensions, and variable bitwidth. Weights and activations use the same bitwidth and the accumulator width is set to $\log_2(9m) \cdot b_w \cdot b_a$.

### 2.3. Communication

Another important aspect of hardware implementation of CNN accelerators is memory communication. The transmission of data from the memory and back is often overlooked by hardware implementation papers [20,25,37] that focus on the raw calculation ability in order to determine the performance of their hardware. In many cases, there is a difference between the calculated performance and real-life performance, since real-life implementations of accelerators are often memory-bound [14,38,39].

For each layer, the total memory bandwidth is the sum of the activation and weight sizes that are read and written from memory. In typical CNNs used, e.g., in vision tasks, the first layers consume most of their bandwidth for activations, whereas in deeper layers that have smaller but higher-dimensional feature maps (and, consequently, a greater number of kernels), weights are the main source of memory communication.

We assume that each PE can calculate one convolution result per clock cycle and the resulting partial sum is saved in the cache. In Figure 5, we present a typical memory access progress at the beginning of the convolutional layer calculation. In the first stage, the weights and first $k$ rows of the activations are read from memory at maximal possible speed, in order to start the calculations as soon as possible. After the initial data are loaded, the unit reaches a "steady state", in which it needs to read, from the memory, only one new input value per clock cycle (other values are already stored in the cache). We assume that the processed signals are two-dimensional (images), which further requires $k$ new values to be loaded at the beginning of each new row.

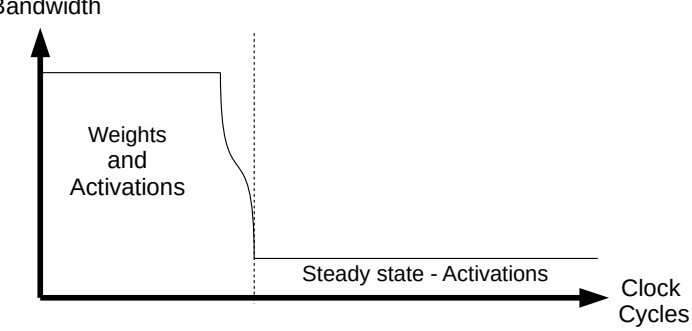

**Figure 5.** Per-layer memory access pattern.

Note that, until the weights and the first activations are loaded, no calculations can be performed. The pre-fetch stage's overhead bandwidth can be mitigated by doing work in larger batch sizes, loading the weights once, and reading several inputs for the same weights. Therefore, we minimize the penalty for reading the weights as compared to reading the actual input data to perform the calculation. However, in the case of real-time processing, larger batches are less feasible, because the stream of data needs to be computed on-the-fly. An alternative solution is to use local memory (SRAM) for storing partial results, weights, and fetched activations for the next cycle of input data. However, the required amount of SRAM may not be available or significantly limit the amount of PEs that can be placed on the same silicon area. This problem is especially relevant in the IPs used as a part of SoCs, which have a highly limited amount of area. The trade-off of available area versus the achievable performance is one of the most important issues in SoC design.

### 2.4. Local Memory

Local memory is used in order to store the weights or activations for further reuse instead of fetching them repeatedly from the main memory. Recently, accelerator designers started to increase the amount of local memory [17,18] in order to fit all of the parameters and avoid using external memory during inference. While this allows for us to avoid memory bottlenecks, local SRAM usage requires a significant amount of resources. Nevertheless, even small amounts of SRAM can be used as cache to reduce the effective

memory bandwidth by storing some input data. The trade-off between the area allocated to local SRAM area and to PEs should be carefully evaluated for each design. SRAM area is almost linear to the amount of bits used, with goodness of fit $R^2 = 0.998$ and $R^2 = 0.916$ for single-port RAM and dual-port RAM, respectively, as shown in Figure 6. Thus, this linear relation can be used in order to accurately estimate the local SRAM area just from the number of bits.

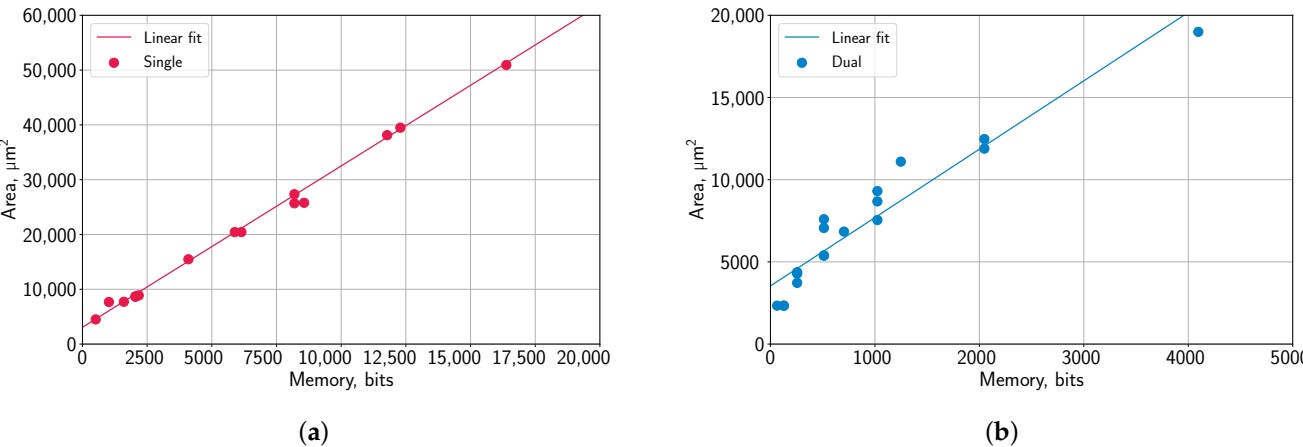

(**a**)         (**b**)

**Figure 6.** SRAM area as a function of memory bits. The data was taken from Synopsys 28 nm Educational Design Kit SRAM specifications. (**a**) Single-port RAM area ($A$) vs. amount of data bits ($B$). The linear fit is $A = 2.94B + 3065$ with a goodness of fit $R^2 = 0.986$. (**b**) Dual-port RAM area ($A$) vs. amount of data bits ($B$). The linear fit is $A = 4.16B + 3535$ with a goodness of fit $R^2 = 0.916$.

In order to compute the complete area of the accelerator, we can use the BOPS metric to scale up our micro-design of the data path to obtain the area of the PEs and derive the area coefficient for the BOPS, $A_D$. For the SRAM, we can derive the relation $A_M$ between the area and the number of bits from Figure 6 to construct the complete area equation:

$$\text{Area} = A_D \cdot BOPS + A_M \cdot B_{\text{SRAM}} + B_D + B_M \tag{2}$$

where $B_{\text{SRAM}}$ is the number of SRAM bits, and $B_D$ and $B_M$ are the free constants.

### 2.5. Roofline Analysis

So far, we discussed the use of BOPS for the prediction of the physical parameters of the final product, such as the expected power and area. In this section, we extend the BOPS model to the system level, by introducing the OPS-based roofline model. The traditional roofline model, as introduced by Williams et al. [30], suggests depicting the dependencies between the performance (e.g., FLOPS/second) and the operation density (the average number of operations per information unit transferred over the memory bus). For each machine, we can draw "roofs": the horizontal line that represents its computational bounds and the diagonal line that represents its maximal memory bandwidth. Figure 7 shows an example of the roofline for three applications assuming infinite compute resources and memory bandwidth. The maximum performance a machine can achieve for any application is visualized by the area below both bounds, shaded in green.

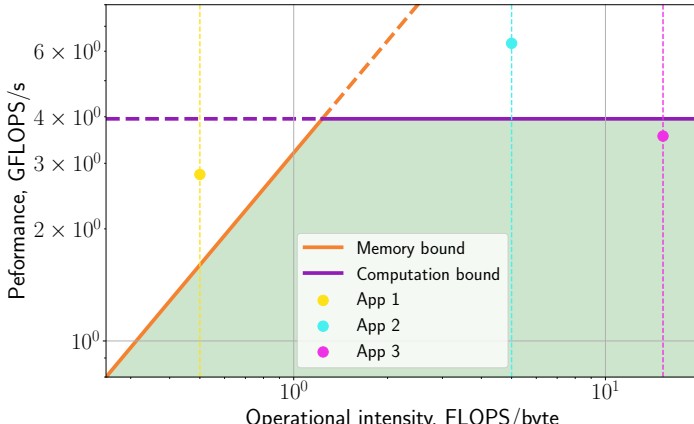

**Figure 7.** Roofline example. In the case of App1, memory bandwidth prevents the program from achieving its expected performance. In the case of App2, the same happens due to limited computational resources. Finally, App3 represents a program that could achieve its maximum performance on a given system.

FLOPS cannot be used for efficient estimation of the complexity of quantized CNNs, as indicated in Section 2.1. Therefore, we introduce a new model that is based on the BOPS metric presented in Section 2.2. This model, to which we refer as the OPS-based roofline model, replaces the GFLOPS/s axis of the roofline plot with a performance metric that is more appropriate for NNs, e.g., the number of operations per second (OPS/s) and the second metric that measures the computational complexity with operations per bit (OPS/bit). Using generic operations and bits allows for us to plot quantized accelerators with different bitwidths on the same plot.

Roofline Analysis Examples

In order to demonstrate the proposed approach, we use two different ResNet-18 layers (a late layer, which is computationally-intensive, and an early one, which is memory-intensive) on four different accelerator designs: 32-bit floating-point, 32-bit fixed-point, and quantized 8-bit and 4-bit fixed-point. The accelerators were implemented while using standard ASIC design tools, as detailed in Section 3 and built using TSMC 28 nm technology, while using standard 2.4 GHz DDR-4 memory with a 64-bit data bus.

The first example employs an accelerator with a silicon area of 1 mm$^2$ and 800 MHz clock speed. The task is the 11th layer of ResNet-18, which has a $3 \times 3$ kernel and 256 input and output features of dimension $14 \times 14$ each. Looking at Table 1, each floating-point multiplier takes 11,786 μm. Thus, in 1 mm$^2$, we can fit

$$\frac{1 \text{ mm}^2}{11,786 \text{ μm}^2} = 84.85 \approx 85 \text{ multipliers,} \tag{3}$$

which, since each PE includes nine multipliers, amounts to

$$\frac{85}{9} = 9.44 \approx 9 \text{ PEs.} \tag{4}$$

With the calculations that are shown in Equations (3) and (4) we can estimate the number of PEs that can be placed on the silicon. Table 2 summarizes the results for different bitwidths, calculated while using data from Figure 1.

**Table 2.** Number of PEs with different bitwidths on 1 mm$^2$ of silicon. Each PE can perform $3 \times 3$ kernel multiplications.

|  | 32-Bit Float | 32-Bit Fixed | 16-Bit Quant. | 8-Bit Quant. |
|---|---|---|---|---|
| PEs | 9 | 60 | 220 | 683 |

In order to calculate the amount of OPS/s required by the layer, under the assumption that a full single pixel is produced every clock, we need to multiply the amount of MAC operations required to calculate one output pixel ($n \times m \times (k^2 + 1)$) by the accelerator frequency. To calculate the OPS/bit for each design, we divide the amount of MAC operations in the layer by the total number of bits transferred over the memory bus, which includes the weights, the input, and the output activations. The layer requires 524.29 TOPS/s to be calculated without stalling for memory access and computation. Table 3 summarizes the available performance of the accelerators and visualized while using the proposed OPS-based roofline analysis shown in Figure 8.

**Table 3.** The amount of computation (OPS/s) provided by the accelerators and memory throughput (OPS/bit) required by the 11th layer of ResNet-18.

|  | 32-Bit Float | 32-Bit Fixed | 16-Bit Quant. | 8-Bit Quant. |
|---|---|---|---|---|
| GOPS/s | 72.00 | 392.0 | 1568 | 5408 |
| OPS/bit | 5.82 | 5.82 | 11.63 | 23.26 |

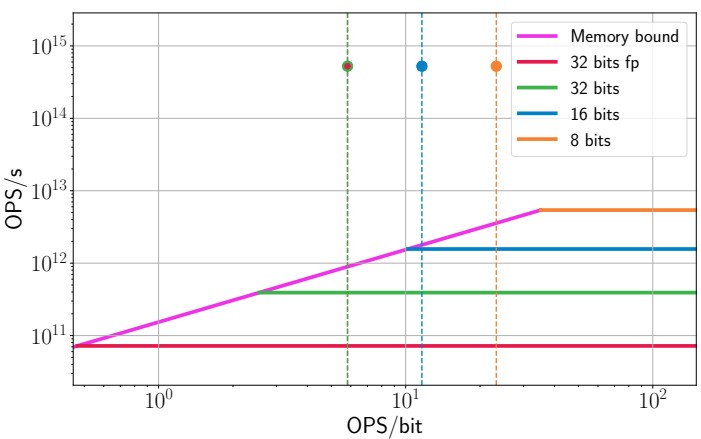

**Figure 8.** OPS roofline: $3 \times 3$ kernel, input and output have 256 features of $14 \times 14$ pixels, 1 mm$^2$ accelerator with an 800-MHz frequency, and a DDR of 2.4 GHz with 64-bit data bus.

In this example, the application's requirements are beyond the scope of the product definition. On one hand, all of the accelerators are computationally bound (all of the horizontal lines are below the application's requirements), indicating that we do not have enough PEs to calculate the layer in one run. On the other hand, even if we decide to increase the computational density by using stronger quantization or by increasing the silicon area (and the cost of the accelerator), we would still hit the memory bound (represented by the diagonal line). In this case, the solution should be found at the algorithmic level or by changing the product's targets, e.g., calculating the layer in parts, increasing the silicon area while decreasing the frequency in order not to hit memory wall, or using another algorithm.

Our second example explores the feasibility of implementing the second layer of ResNet-18 that has a $3 \times 3$ kernel and 64 input and output features of dimension $56 \times 56$. For this example, we increase the silicon area to 6mm$^2$ and lower the frequency to 100 MHz,

as proposed earlier, and then add a 4-bit quantized accelerator for comparison purposes. The layer requires 4.1 $^{\text{GOPS}}$/s. Table 4 summarizes the accelerators results and they are visualized with the OPS-based roofline analysis shown in Figure 9.

**Table 4.** The amount of computation (OPS/s) provided by the accelerators and memory throughput (OPS/bit) required by the second layer of ResNet-18.

|  | 32-Bit Float | 32-Bit Fixed | 16-Bit Quant. | 8-Bit Quant. | 4-Bit Quant. |
|---|---|---|---|---|---|
| GOPS/s | 49.00 | 324.0 | 1296 | 3969 | 11,236 |
| OPS/bit | 9.16 | 9.16 | 18.32 | 36.64 | 73.27 |

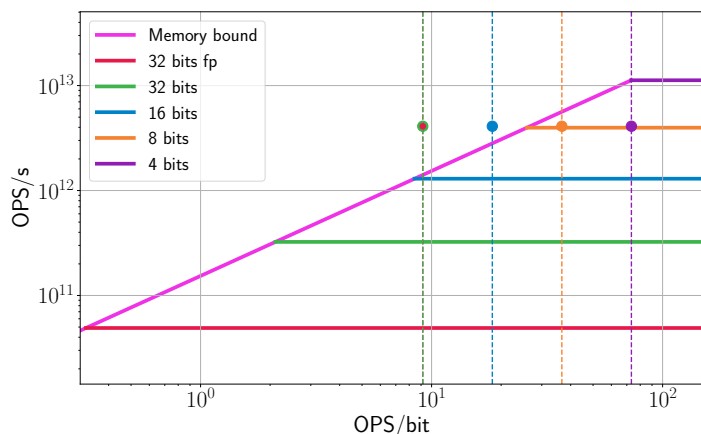

**Figure 9.** OPS roofline: $3 \times 3$ kernel, input and output have 64 features of $56 \times 56$ pixels, 6 mm$^2$ accelerator with with an 100-MHz frequency, and a DDR of 2.4 GHz with 64-bit data bus.

From Figure 9. we can see that our 32-bit and 16-bit accelerators are still computationally bound, while the 8-bit and 4-bit quantized accelerators meet the demands of the layer. In particular, the 8-bit accelerator is located at the edge of the computational ability, which means that this solution has nearly optimal resource allocation, since the hardware is fully utilized. Still, the final choice of the configuration depends on other parameters, such as the accuracy of the CNN.

Both of the examples demonstrate that decisions made at early stages have a critical impact on the quality of the final product. For example, applying aggressive quantization to the network or increasing the silicon size may not improve the overall performance of the chip if it is bounded by memory. From the architect's point of view, it is important to balance computation and data transfer. Nonetheless, this balance can be achieved in different ways: at the micro-architecture level, at the algorithmic level, or by changing the data representation. The architect may also consider: (1) changing the hardware to provide faster communication (which requires more power and is more expensive), (2) applying communication bandwidth compression algorithms [40,41], (3) using fewer bits to represent weights and activations (using 3- or 4-bit representation may solve the communication problem, at the cost of reducing the expected accuracy), or (4) changing the algorithm to transfer the data slower (even though that solves the bandwidth issue, the possible drawback is reduced throughput of the whole system). The proposed OPS-based roofline model helps the architect to choose between alternatives. After making major architectural decisions, we can use BOPS in order to estimate the impact of different design choices on the final product, such as the expected area, power, optimal operational point, etc.

The next section examines these design processes from the system design point of view.

## 3. Results

In this section, we show how the proposed method can be used as an estimator for area changes in the early design stage. We conducted an extensive evaluation of the design and implementation of a commonly used CNN architecture for ImageNet [42] classification, ResNet-18 [43]. We also show an evaluation on existing hardware [20] whle using our roofline model, and show how we can predict performance bottlenecks on a particular VGG-16 implementation.

### 3.1. Experimental Methodology

We start the evaluation with a review of the use of BOPS as part of the design and implementation process of a CNN accelerator. This section shows the trade-offs that are involved in the process and verifies the accuracy of the proposed model. Because PEs are directly affected by the quantization process, we focus here on the implementation of a single PE. The area of an individual PE depends on the chosen bitwidth, while the change in the amount of input and output features changes both the required number of PEs and size of the accumulator. In order to verify our model, we implemented a weight stationary CNN accelerator, which reads the input feature for each set of read weights and can calculate *n* input features and *m* output features in parallel, as depicted in Figure 10. For simplicity, we choose an equal number of input and output features. In this architecture, all of the input features are routed to each of the *m* blocks of the PEs, each calculating a single output feature.

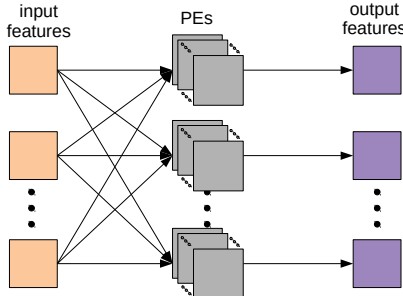

**Figure 10.** All-to-all topology with $n \times m$ processing elements.

The implementation was done for an ASIC while using the TSMC 28 nm technology library, an 800 MHz system clock, and in the nominal corner of $V_{DD} = 0.81$ V. We used the value of 0.2 for the power analysis, input activity factor, and sequential activity factor. Table 5 lists the tool versions.

**Table 5.** Computer-Aided Design (CAD) Design Tools.

| | |
|---|---|
| Language | Verilog HDL |
| Logic Simulation | ModelSim 19.1 |
| Synthesis | Synopsys Design Compiler 2017.09-SP3 |
| Place and route | Cadence Innovus 2019.11 |

For brevity, we only present the results of experiments at the 800-MHz clock frequency. We performed additional experiments at 600 MHz and 400 MHz. Because the main effect of changing the frequency is reduced power usage and not the area of the cells (the same cells that work for 800 MHz will work at 600 MHz, but not the other way around), we do not show these results. Lowering the frequency of the design can help to avoid the memory bound, but incurs the penalty of longer runtime, as shown in Section 2.5.

Our results show a high correlation between the design area and BOPS. The choice of an all-to-all topology that is shown in Figure 10 was made because of an intuitive understanding of how the accelerator calculates the network outputs. However, this choice

has a greater impact on the layout's routing complexity, with various alternatives incuding broadcast or systolic topologies [16]. For example, a systolic topology, which is a popular choice for high-end NN accelerators [14], eases the routing complexity by using a mesh architecture. Although it reduces the routing effort and improves the flexibility of the input/output feature count, it requires more complex control for the data movement to the PEs.

In order to verify the applicability of BOPS to different topologies, we also implemented a systolic array that is shown in Figure 11, where each PE is connected to four neighbors with the ability to bypass any input to any output without calculations. The input feature accumulator is located at the input of the PE. This topology generates natural $4 \times 1$ PEs, but. with proper control, it is possible to create flexible accelerators.

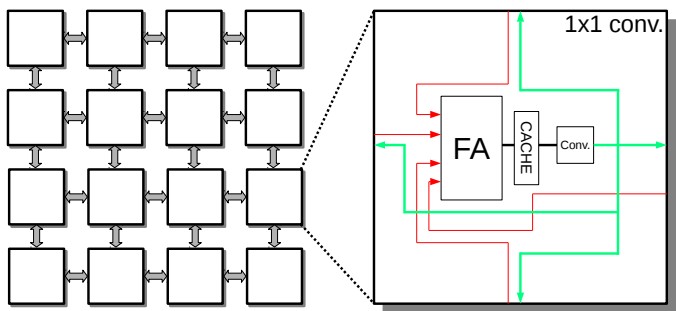

**Figure 11.** Systolic array of PEs.

In the systolic design, we generated three square arrays of $4 \times 4$, $8 \times 8$, and $16 \times 16$ PEs, with $b_w = b_a \in \{4, 6\}$. The systolic array area was found to be in linear relation with BOPS, with the goodness of fit $R^2 = 0.9752$, as shown in Figure 12.

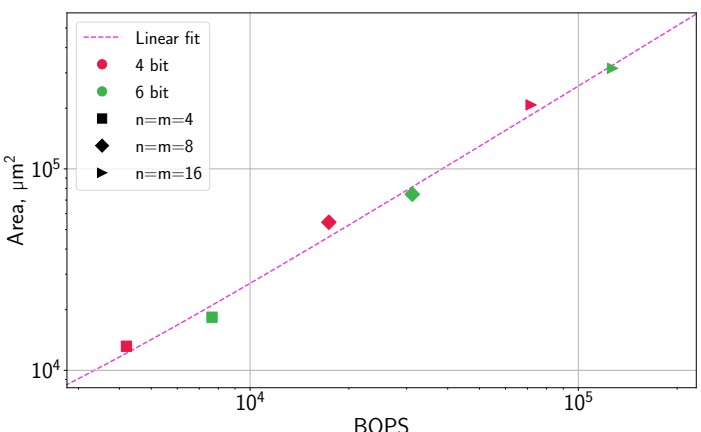

**Figure 12.** Area ($A$) vs. BOPS ($B$) for a systolic array of $3 \times 3$ PEs with variable input (n) and output (m) feature dimensions, and variable bitwidth. Weights and activations use the same bitwidth and the accumulator width is set to $\log_2(9m) \cdot b_w \cdot b_a$.

### 3.2. System-Level Design Methodology

In this section, we analyze the acceleration of ResNet-18 while using the proposed metrics and show the workflow for the early estimation of the hardware cost when designing an accelerator. We start the discussion by targeting an ASIC that runs at 800 MHz, with $16 \times 16$ PEs and the same 2.4 GHz DDR-4 memory with a 64-bit data bus, as used in Section 2.5. The impact of changing these constraints is discussed at the end of the section. For the first layer, we replace the $7 \times 7$ convolution with three $3 \times 3$ convolutions, as proposed

by He et al. [44]. This allows for us to simplify the analysis by employing universal $3 \times 3$ kernel PEs for all layers.

We start the design process by comparing different alternatives while using the new proposed OPS-based-roofline analysis, since it helps to explore the design trade-offs between the multiple solutions. We calculate the amount of OPS/s provided by $16 \times 16$ PEs at 800 MHz and the requirements of each layer. In order to acquire the roofline, we need to calculate the OPS/bit, which depend on the quantization level. For ResNet-18, the current state-of-the-art [45] achieves 69.56% top-1 accuracy on ImageNet for 4-bit weights and activations, which is only 0.34% less than the 32-bit floating-point baseline (69.9%). Thus, we decided to focus on 2-, 3-, and 4-bit quantization both for weights and activations, which can achieve 65.17%, 68.66%, and 69.56% top-1 accuracy, correspondingly.

For a given bitwidth, the OPS/bit is calculated by dividing the total number of operations by the total number of bits transferred over the memory bus, consisting of reading weights and input activations and writing output activations. Figure 13 presents the OPS-based roofline for each quantization bitwidth. Note that, for each layer, we provided two points: the red dots are the performance required by the layer, and the green dots are the equivalent performance while using partial-sum computation.

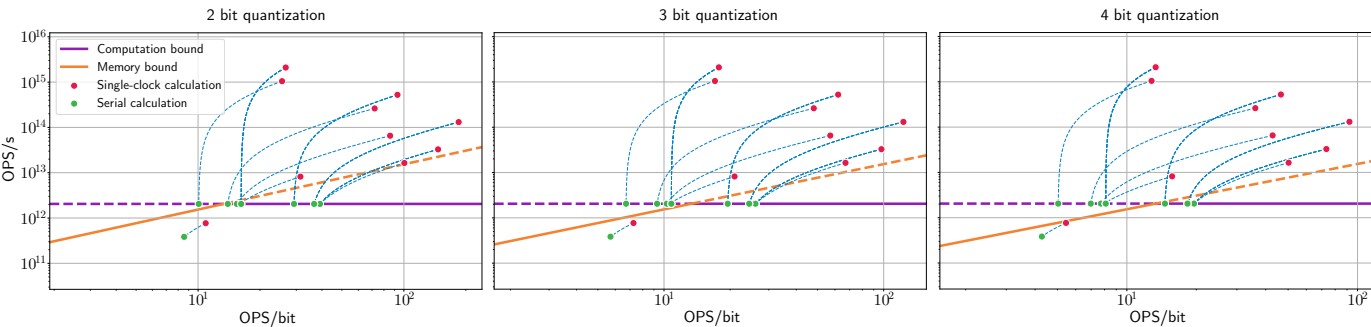

**Figure 13.** ResNet-18 roofline analysis for all layers. Red dots are the performance required by the layer, and green dots are the equivalent performance using partial-sum computation. The blue curves connect points corresponding to the same layer and they are only displayed for convenience.

Figure 13 indicates that this accelerator is severely limited by both computational resources and a lack of bandwidth. The system is computationally bounded, which could be inferred from the fact that it does not have enough PEs to calculate all of the features simultaneously. Nevertheless, the system is also memory-bound for any quantization level, which means that adding more PE resources would not fully solve the problem. It is crucial to make this observation at the early stages of the design: it means that micro-architecture changes would not be sufficient to obtain optimal performance.

One possible solution, as mentioned in Section 2.5, is to divide the channels of the input and output feature maps into smaller groups, and use more than one clock cycle in order to calculate each pixel. In this way, the effective amount of the OPS/s required for the layer is reduced. When the number of feature maps is divisible by the number of available PEs, the layer will fully utilize the computational resources, which is the case for every layer except the first one. However, reducing the number of PEs also reduces the data efficiency and, thus, the OPS/bit also decreases, shifting the points to the left on the roofline plot.

Thus, some layers still require more bandwidth than the memory can supply. In particular, in the case of 4-bit quantization, most of the layers are memory bounded. The only option that properly utilizes the hardware is 2-bit quantization, for which all of the layers except one are within the accelerator's memory bound. If the accuracy for 2-bit quantized network is insufficient and finer quantization is required, then it is possible to reallocate some of the area used for the PEs to be used for additional local SRAM. By caching the activations and output results for the next layer, we can reduce the required

bandwidth from external memory at the expense of performance (i.e., increasing total inference time). Reducing the PE count lowers the compute bound on the roofline, but, at the same time, the use of SRAM increases operation density (i.e., moves the green dots in Figure 13 to the right), possibly within hardware capabilities. Alternative solutions for the memory-bound problem include changing the CNN architecture (for example, using smaller amount of wide layers [46]), or adding a data compression scheme on the way to and from the memory [40,41,47].

At this point, BOPS can be used in order to estimate the power and area of each alternative for implementing the the accelerator while using the PE micro-design. In addition, other trade-offs can be considered, such as the influence of modifying some parameters that were fixed at the beginning: lowering the ASIC frequency will decrease the computational bound, which reduces the cost and only hurts the performance if the network is not memory bounded. An equivalent alternative is to decrease the number of PEs. Both of the procedures will reduce the power consumption of the accelerator, as well the computational performance. The system architect may also consider changing the parameters of the algorithm, e.g., change the feature sizes, use different quantization for the weights and for the activations, include pruning, etc.

It is also possible to reverse the design order: start with a BOPS estimate of the number of PEs that can fit into a given area, and then calculate the ASIC frequency and memory bandwidth that would allow for full utilization of the accelerator. This can be especially useful if the designer has a specific area or power goal.

To summarize, it is extremely important, from an architectural point of view, to be able to predict in the early stages of the design whether the proposed (micro)architecture is going to meet the project targets. At the project exploration stage, the system architect can choose from multiple alternatives in order to make the right trade-offs (or even negotiate to change the product definition and requirements). Introducing such alternatives later may be very hard or even impossible.

*3.3. Evaluation of Eyeriss Architecture*

In this section, we demonstrate the evaluation of existing CNN hardware architecture—the Eyeris [20] implementation of VGG-16—while using our modified roofline analysis. We visualized the required performance (compute and memory bandwidth) of each layer in Figure 14. As earlier, red dots denote the required performance, the purple horizontal line shows the available compute resource, and the diagonal orange line is the memory bandwidth bound. The required performance is obviously compute bounded since PEs are not enough to calculate all of the layers; the calculation is performed in cycles. The required performance when calculating in cycles is plotted in green dots. If we compare the Eyeriss roofline analysis to our architecture from Section 3, we can see a difference in the movement of the required performance. This phenomenon is the result of the different hardware architectural structures. Our example utilized weight stationary architecture, which has an overhead when calculating in cycles: the input features are read multiple times for each set of weights. Eyeriss architecture uses a row stationary approach and it has enough local memory to re-use all of the weights and the activations before reading additional data. It allows for the overhead of re-reading the activations for each set of weights to be avoided. Because the roofline analysis shows asymptotical performance, data compression and data-drop techniques [20] that may help reduce memory bandwidth and compute requirements are excluded from the roofline analysis. While these approaches can change the hardware requirements, it is infeasible to accurately estimate their impact on the performance, due to their dependency on the data distribution.

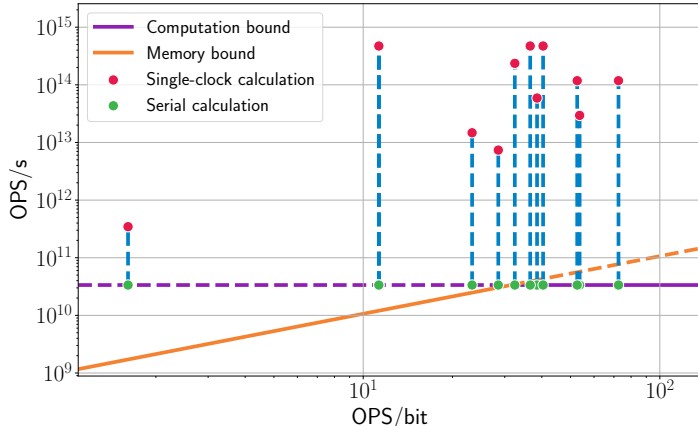

**Figure 14.** VGG-16 on Eyeriss [20] hardware. Red dots are the performance required by the layer, and green dots are the equivalent performance using partial-sum computation. The blue curves connect points corresponding to the same layer and they are only displayed for convenience.

Our analysis shows that VGG-16 on Eyeriss hardware has some memory bounded layers. While two of these layers are close to the memory bound and can possibly get inside the memory bound of the compression scheme [20], the first and the three last layers suffer from poor performance compared to other layers. To evaluate the slowdown, in Table 6 we show the real performance that is based on the roofline model as well as the amount of time that is required for calculations (in the absence of a memory bound). Our prediction of the performance is similar to the performance results that are shown by Chen et al. [20].

**Table 6.** Achievable performance of VGG-16 on Eyeriss hardware as seen from the roofline analysis. The latency is the amount of time the execution units need to calculate the data in that layer.

| Layer | Latency [ms] | Latency from Roofline [ms] |
|---|---|---|
| conv1-1 | 7.7 | 158.9 (+1963.6%) |
| conv1-2 | 165.2 | 191.4 (+15.9%) |
| conv2-1 | 82.6 | 117.3 (+42%) |
| conv2-2 | 165.2 | 165.2 |
| conv3-1 | 82.6 | 82.6 |
| conv3-2 | 165.2 | 165.2 |
| conv3-3 | 165.2 | 165.2 |
| conv4-1 | 82.6 | 84.2 |
| conv4-2 | 165.2 | 165.2 |
| conv4-3 | 165.2 | 165.2 |
| conv5-1 | 41.3 | 120.9 (+192.7%) |
| conv5-2 | 41.3 | 120.9 (+192.7%) |
| conv5-3 | 41.3 | 120.9 (+192.7%) |

In the case of Eyeriss, adding more local SRAM cannot resolve the memory bound issue. Eyeriss already re-uses the weights and activations (i.e., no data are read multiple times), so the only option is to increase the memory speed. To conclude, the roofline analysis results should be a tool for the architect to use during the planning process. Performance degradation in some layers may be tolerable, as long as we have an appropriate metric to accurately evaluate the impact on the entire network. The main benefit of using the roofline analysis is that we can predict the areas where the network architecture is not optimal and where we may need to focus on the design. It is up to the architect of the hardware to make these decisions.

## 4. Discussion

### 4.1. Conclusions

CNN accelerators are commonly used in different systems, starting from IoT and other resource-constrained devices, and ending in datacenters and high-performance computers. Designing accelerators that meet tight constraints is still a challenging task, since the current EDA and design tools do not provide enough information to the architect. To make the right choice, the architects need to understand the impact of their high-level decisions on the final product in the early design stages as well as to be able to make a fair comparison between different design alternatives.

In this paper, we introduced a framework for early-stage performance analysis that works on any quantization level of the operands. We also presented the OPS-based roofline model as a supporting tool for the architect. We showed that our framework allows for a comparison of different design alternatives and an evaluation of the solution's feasibility. Utilizing BOPS [29] as the complexity metric, we can approximate changes in accelerator resource requirements (area and power) that result from various possible architectural changes. We evaluated the proposed method on several examples of realistic designs, including the Eyeriss accelerator [20]. In particular, our analysis framework confirms that CNN accelerators are more likely to be a memory rather than computationally bound [14,39]. We conclude that, by using this analysis framework, architects will be able to optimize their design performance in a fast development cycle.

Although this paper is mainly focused on ASIC-based architectures, the same methodology can be applied to many other systems, including FPGA-based implementations and other system-specific domains that allow for trading-off accuracy and data representation with different physical parameters, such as power, performance, and area.

### 4.2. Future Work

Creating a new hardware performance analysis framework for networks other than CNNs, such as recurrent and graph neural networks, will be a powerful addition to ours. We would like to explore new types of hardware accelerator architecture, such as the "on-the-fly" variable quantization accelerator, where the same hardware elements can be used for low bitwidth quantization layers or combined for high bitwidth quantization layers in the same network. Developing novel machine learning hardware accelerators based on "cache-less" architecture, where the area of the SRAM can be used for placing more PEs on the silicon, and the activations and weights will flow inside between the PEs is an interesting future direction. Such an architecture creates additional difficulties alongside the problems of layout and routing on the silicon, and new algorithms will be needed in order to map the massive PE array's workload.

**Author Contributions:** Conceptualization, A.K., C.B. and A.M.; Formal analysis, A.K., C.B. and E.Z.; Funding acquisition, A.M.B. and A.M.; Investigation, A.K., C.B., E.Z. and Y.Y.; Methodology, A.K., C.B. and Y.Y.; Project administration, A.M.B. and A.M.; Software, A.K. and Y.Y.; Supervision, F.G., A.M.B. and A.M.; Validation, A.K., C.B. and Y.Y.; Visualization, E.Z.; Writing—original draft, A.K., C.B. and E.Z.; Writing—review & editing, F.G., A.M.B. and A.M. All authors have read and agreed to the published version of the manuscript.

**Funding:** The research was funded by the Hyundai Motor Company through the HYUNDAI-TECHNION-KAIST Consortium, National Cyber Security Authority, and the Hiroshi Fujiwara Technion Cyber Security Research Center.

**Data Availability Statement:** No new data were created or analyzed in this study. Data sharing is not applicable to this article.

**Conflicts of Interest:** The authors declare no conflict of interest.

**Abbreviations**

The following abbreviations are used in this manuscript:

| | |
|---|---|
| ASIC | Application-Specific Integrated Circuit |
| CAD | Computer-Aided Design |
| CNN | Convolutional Neural Network |
| DDR | Double Data Rate (Memory) |
| DL | Deep Learning |
| EDA | Electronic Design Automation |
| FLOPS | Floating point Operations |
| FMA | Fused Multiply-Add |
| FPGA | Field Programmable Gate Array |
| GOPS | Giga Operations |
| HDL | Hardware Description Language |
| IC | Integrated Circuit |
| IP | Intellectual Property |
| MAC | Multiply Accumulate |
| NN | Neural Network |
| OPS | Operations |
| PE | Processing Engine |
| RAM | Random Access Memory |
| SoC | System on a Chip |
| SRAM | Static Random Access Memory |
| TOPS | Tera Operations |
| TSMC | Taiwan Semiconductor Manufacturing Company |
| VLSI | Very Large-Scale Integration |

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
