# Peer review of "Early-Stage Neural Network Hardware Performance Analysis"

_sustainability, doi:10.3390/su13020717_

Round 1

Reviewer 1 Report

I like the topic that this paper dines into. It's important to have a well-defined metric to assist architecture-level decisions in the early design stages.

The major issue with the current manuscript is that, it does not clearly point out what exactly the hardware-aware complexity metric (HCM) is. It's nice to have all the examples the authors given in this manuscript. But as a potential user, I cannot tell where to find the definition and equation of HCM. 

In Section 2.3, the authors introduce BOPS. It has a clear definition and its equation is provided. I would expect to see the same for HCM. 

Also, the evaluation of the HCM needs to be revised accordingly. For example, in Section 3.2 and 3.3, I cannot see any "HCM" keyword rather than the section title. Let's say if the "area" is a part of HCM, then using HCM-area can make it more clear for readers to understand. 

"This paper proposes to use a different metric for assessing the complexity of CNN-based architectures: the number of bit operations (BOPS) as defined by Baskin et al. [25]." This claim is confusing. BOPS was proposed in another paper. It's not a new metric proposed in this paper. The authors need to make it clear, what's new here. If BOPS was used to assess A in previous work, while BOPS is used to assess B in this work, you need to make these things clear.

In addition, the relationship between HCM and BOPS is not clear to me. It's back to my question mentioned above. What's HCM? What's the role of BOPS? Is it part of your HCM?

Reviewer 2 Report

This paper is mainly centered on a hardware-aware complexity metric to identify bottlenecks in the DNN hardware design. The authors demonstrated how the proposed metric can help evaluate different architecture alternatives of resource-restricted NN accelerators (e.g., part of real-time embedded systems) and avoid making design mistakes at early development stages. This research work is interesting for determining the shape and size of the personal space of a human when passed by a robot. However, this paper has several limitations and the standard is not enough, and address the following items would result in a good paper.

  1. The introduction is not sufficient. When it comes to the hardware-aware complexity metric for neural network architectures, the machine learning algorithm is widely used in related works. In the literature analysis, it is recommended to read the following works and consider their similar applications in the introduction and future works. For example, improved recurrent neural network-based manipulator control with the remote center of motion constraints: experimental results; an incremental learning framework for human-like redundancy optimization of anthropomorphic manipulators; a smartphone-based adaptive recognition and real-time monitoring system for human activities; deep neural network approach in robot tool dynamics identification for bilateral teleoperation.

  1. To illustrate the advanced performance of the proposed method, what is the main task to be solved in this paper? It is recommended to present in the first section so that it can highlight the specific scope of this article. For example, what can the hardware-aware complexity metric solve for bottlenecks in the DNN hardware design?

  1. In Section 1, the authors present the proposed 3 × 3 kernels 8-bit processing engine (PE) layout using the TSMC 28nm technology in fig.1. It is recommended to put this fig in section 2. Table 1 shows the key characteristics of 32-bit floating-point and 32-bit fixed-point multiplier designs. However, there is no corresponding explanation to the table, which makes this part hard to read.

  1. 2 exhibits the silicon area of the PE as a function of the bandwidth, and the authors performed a polynomial regression and observed a quadratic dependence of the PE area on the bandwidth, with the coefficient of determination R2 = 0.9999877. However, there is no polynomial regression formulation that corresponds to Fig.2.

  1. In Fig. 3, the authors calculated BOPS values for the PEs from Fig. 2 and plotted them against the area. And Fig. 4 shows that the area depends linearly on the BOPS for the range of two orders of magnitude of total area with the goodness of fit R = 0.9980. However, the values of "R" shown in the two figures are different. The authors should give some explanation on the connection between the two figures.

  1. Two cases are taken into consideration for demonstrating the performance of the proposed approaches, which is the BOPS's use as a metric for the hardware complexity of neural networks algorithms and the use of the OPS-based roofline model. However, before giving the content of the experiment, the steps and initial conditions of the experiment should be clearly explained. And the authors should focus on their method more.

  1. To let readers better understand future work, please give specific research directions.

  1. There is some mistake of grammar and English expression in this paper should be considered, for example, "there has been a tremendous growth", "growing with faster pace", "which is inspired from". Please check the overall paper carefully. And revise the expression seriously.

Round 2

Reviewer 1 Report

The authors have addressed all my concerns.

Reviewer 2 Report

This paper has addressed all of my concens. No more revision is required from my side. The current version can be accepted.